# Evaluation of spatial Bayesian Empirical Likelihood models in analysis of small area data

**Farzana Jahan** *, **Daniel W. Kennedy**, **Earl W. Duncan** , **Kerrie L. Mengersen**

School of Mathematical Sciences, ARC Centre of Excellence in Mathematical and Statistical Frontiers (ACEMS), QUT Centre for Data Science, Faculty of Science, Queensland University of Technology, Brisbane, Queensland, Australia

* fjahan0518@gmail.com

## Abstract

Bayesian empirical likelihood (BEL) models are becoming increasingly popular as an attractive alternative to fully parametric models. However, they have only recently been applied to spatial data analysis for small area estimation. This study considers the development of spatial BEL models using two popular conditional autoregressive (CAR) priors, namely BYM and Leroux priors. The performance of the proposed models is compared with their parametric counterparts and with existing spatial BEL models using independent Gaussian priors and generalised Moran basis priors. The models are applied to two benchmark spatial datasets, simulation study and COVID-19 data. The results indicate promising opportunities for these models to capture new insights into spatial data. Specifically, the spatial BEL models outperform the parametric spatial models when the underlying distributional assumptions of data appear to be violated.

## 1 Introduction

Bayesian Empirical Likelihood (BEL) was first described by Lazar [1] based on early work on Empirical Likelihood (EL) by Owen. [2]. The concept of EL has been utilised in Bayesian analysis in a few instances since then [3–5]. As discussed in section 2.1, BEL provides a flexible semi-parametric approach using data to approximate the likelihood part of the Bayesian posterior combined with parametric prior distributions.

In the context of spatially dependent data, an EL framework has been developed in a frequentist set-up by [6–9]. BEL semi-parametric approaches for modelling areal spatial data were introduced by Chaudhuri and Ghosh [10] and Porter et al. [11, 12] extending small area estimation models [13]. These BEL spatial approaches utilise a Bayesian hierarchical framework, moving the spatial dependence structure to the parameter model of the hierarchy and applying EL at the observation level [11]. Chaudhuri and Ghosh [10] introduced area-level and unit-level models which can handle both discrete and continuous data, with informative priors for the spatial random effects as independent Gaussian and Dirichlet process mixture priors. Porter et al. [11] provided a more complete version of this model and suggested that it

**Data Availability Statement:** All the datasets used in this article are publicly accessible data and are free to download for anyone interested. No special privileges are needed to access the datasets, and the authors also did not have any special privileges

to access the data for analysis in the current article. The links to download the datasets used are as follows: 1. Scottish Lip Cancer Data: https:// geodacenter.github.io/data-and-lab/scotlip/ 2. North Carolina SIDS data: https://r-spatial.github. io/spdep/articles/sids.html 3. COVID19 data: https://github.com/owid/covid-19-data/tree/ master/public.

**Funding:** This research was supported by an ARC Australian Laureate Fellowship for project, Bayesian Learning for Decision Making in the Big Data Era under Grant no. FL150100150, awarded to KM. The funders had no role in study design, data collection and analysis, decision to publish, or preparation of the manuscript.

**Competing interests:** The authors have declared that no competing interests exist.

could be generalised to spatial-temporal dependencies. Porter et al. [11] formulated a different set of lattice priors utilising the generalised Moran basis [14] for the spatial dependencies. The multivariate extension of the Bayesian semi-parametric hierarchical empirical likelihood (BSHEL) model of [11] can be found in Porter et al. [12].

The BEL models described above focused on a selected set of priors to represent spatial dependence. However, a class of very popular spatial priors, namely the conditional autoregressive (CAR) priors for the spatial random effects in areal data analysis [15, 16] have still not been explored in the BEL framework. This article aims to address this research gap by formulating BEL semi-parametric spatial models applying CAR structure priors for spatial random effects.

The CAR priors have gained in popularity for modelling spatial data and have been employed in many applications of spatial modelling, such as disease mapping [17]. Besag et al. [18] introduced the CAR prior, which has become known as the BYM prior. There are now many variants of this prior and corresponding model such as the Cressie model [19], the Leroux model [20] and the Lu model [21]. For a detailed comparison of different CAR priors in spatial analysis under Bayesian parametric framework, see Lee, D. [15] and Rampaso et al. [16]. The Leroux model has been termed as a flexible CAR structure for modelling spatial random effects, since it consists of a spatial dependence parameter ($\rho$) taking different values according to the underlying spatial autocorrelation present in the data. Special cases of the Leroux model give rise to independent Gaussian (IG) priors for spatial random effects, when no spatial structure is needed for modelling areal data ($\rho = 0$) and intrinsic conditional autoregressive (ICAR) priors ($\rho = 1$) (which is the spatial prior considered in the BYM model [18]).

The present study develops spatial BEL (SBEL) models for the two popular CAR prior choices, Leroux and BYM. The proposed SBEL-CAR models are illustrated by analysing areal data on an irregular lattice using two benchmark examples on Scottish lip cancer [22] and North Carolina Sudden Infant Death Syndrome (SIDS) [23] and on a very recent example using a COVID-19 dataset for Europe. The performance of the proposed models and other existing spatial BEL models and their parametric counterparts are compared. The models are also illustrated using simulated datasets.

The development of the proposed models is achieved by extending the Bayesian semiparametric hierarchical empirical likelihood (BSHEL) model proposed by Porter et al. [11]. A recap of the BEL models, BEL spatial models and Bayesian parametric spatial models is given in Section 2. Section 3 contains the formulation of SBEL-CAR models with an algorithm to obtain the posterior samples of interest. The application of the SBEL-CAR models to case study datasets and simulated datasets is reported in section 4 followed by a final discussion in section 5.

## 2 Background on BEL and SBEL models

This section briefly discusses the background of Bayesian Empirical Likelihood (BEL) and SBEL models from the existing literature.

### 2.1 Recap of Bayesian Empirical Likelihood

**2.1.1 Empirical likelihood.** Empirical likelihood (EL) combines the reliability of nonparametric methods with the flexibility and effectiveness of the likelihood approach. To overcome the model misspecification and lack of robustness of parametric likelihood, a non parametric analogue of parametric likelihood, empirical likelihood (EL) was introduced by Owen [2]. Owen [2] initially showed that the empirical likelihood ratio function can be used to construct confidence intervals for a sample mean, for a class of M-estimates and for other

differentiable statistical functional. These methods were framed as a non-parametric extension of Wilk's [24] theorem for parametric likelihood ratio tests. Later EL was expanded to all types of estimating equations by Qin and Lawless [25].

For data points $y_1, y_2, \ldots, y_n$ from some unknown distribution $F$, let some functional of $F$, say $\theta(F)$ be the parameter of interest for inference, which can be determined by the estimating equation $f(y_i, \theta)$, the EL function can be defined as [26]:

$$\hat{L}(F, w) = \prod_{i=1}^{n} w_i$$

where $w_i$ satisfies the constraints $\sum_{i=1}^{n} w_i = 1, \sum_{i=1}^{n} w_i f_i(y_i, \theta) = 0$.

Constrained optimisation of the EL ratio function is carried out in order to obtain the EL weights $w_i$ which are used as the data likelihood [26].

The introduction to estimating equations to EL enhanced the scope of EL to so many different applications including generalised linear models [27], time series [7, 28], econometrics [29], regression analysis [30, 31], survival analysis [32] and many more. For more details on scope and benefits of EL, we refer the reader to Lazar [33], which reviewed EL from its initiation to current developments in theory and applications along with the potential for future development.

**2.1.2 Bayesian Empirical Likelihood.** In a Bayesian framework, the likelihood is used to update a prior distribution and yield posterior inference. Lazar [1] argued that EL can be used in place of a density and, when multiplied by the prior of the parameter of interest can yield the posterior distribution in such an analysis. The author explored the characteristics of Bayesian inference using EL instead of a parametric density. Starting from a well-known parametric case of a Bayesian posterior of a parameter vector $\theta$, Schennach [3] derived an EL posterior using the weights attributed to the sample points, which can be calculated by solving an entropy maximisation problem,

$$p(\theta|Y) \propto p(\theta) \prod_{i=1}^{n} w_i^*(\theta) \qquad (1)$$

where $p(\theta)$ is a given prior on $\theta$ and $(w_1^*(\theta), \ldots, w_n^*(\theta))$ is the solution of:

$$argmax \sum_{i=1}^{n} - w_i \log(w_i), \text{subject to} \sum_{i=1}^{n} w_i = 1, \sum_{i=1}^{n} w_i f_i(y_i, \theta) = 0 \qquad (2)$$

where $f_i(y_i, \theta)$ represents the estimating equations of interest. The estimating equations can be formulated using the moment conditions [3, 5].

Schennach [3] showed that for large enough samples, BEL offered similar results to Bayesian bootstrap. Grendar and Judge [34] demonstrated that the BEL is an asymptotic approximation of the Bayesian maximum a posteriori probability estimators, which provided additional justification of using EL in a Bayesian setting [33]. As a result, BEL has been applied in quantile regression [35], ridge and lasso regression [36], inference with complex survey data [37], spatial data analysis [10–12, 38] and so on. The spatial analysis using BEL is the focus of this study.

## 2.2 Recap of spatial BEL models

BEL has been employed for spatial analysis [10–12] and has been found to provide precise estimation of small area effects. Chaudhuri and Ghosh [10] introduced area-level and unit-level models using BEL by extending the traditional Fay-Herriot (FH) model [13] using an EL

framework. Following this work, Porter et al. [11] provided a general hierarchical Bayesian framework incorporating empirical likelihood methodology for the data model and developed a spatial FH model used for small area estimation (SAE). The FH model for SAE can be written as:

$$Y_i = \mu_i + \epsilon_i \tag{3}$$

$$\mu_i = X'_i \boldsymbol{\beta} + \psi_i \tag{4}$$

where $Y_i$, $i = 1, 2, \ldots, n$, is a design unbiased estimate of $\mu_i$, and $\epsilon_i$ is an unstructured error component, $X_i$ is the vector of covariate information for area $i$, $\boldsymbol{\beta}$ is the vector of fixed covariate effects, $\boldsymbol{\beta} = (\beta_0, \beta_1, \ldots, \beta_p)'$ and $\psi_i$ is spatially referenced random effect for area $i$.

Three forms of priors for the vector of spatial random effects $\boldsymbol{\psi}$ were specified by these authors. [10] suggested two options, the first being an independent and identically distributed (iid) Gaussian distribution (IG) with variance $A$ following a Inverse Gamma distribution

$$\psi_i \sim N(0, A), A \sim InverseGamma(\alpha_1, \alpha_2).$$

The second option was a Dirichlet process (DP) with a Gaussian base,

$$\psi_i | G \sim G, G | A \sim DP(\alpha, \mathcal{G})$$

where $DP(\alpha, \mathcal{G})$ represents a Dirichlet process with precision parameter $\alpha$ and a base measure $G_0 \sim N(0, A)$

In contrast, Porter et al. [11] suggested the use of a generalised Moran basis [14],

$$\pi(\boldsymbol{\psi}) \propto \tau^{q/2} \exp\left\{ -\frac{1}{2} \tau \boldsymbol{\psi}^{*'} M'(B_+ - B) M \boldsymbol{\psi}^* \right\}$$

where, $B$ is an adjacency matrix for a first order Intrinsic Gaussian Markov Random Field (IGMRF) (rank $(B) = n - 1$), $B_+$ is a diagonal matrix with $\{B_+\}_{i,i} = \Sigma_{j \in ne(i)} b_{ij}$, $b_{ij} = 1$ if i and j are adjacent and 0 otherwise, where $j \in ne(i)$ means that area $j$ is a neighbour of area $i$. The vector $M$ is the set of eigen-vectors corresponding to the non-zero eigenvalues of the matrix $P_c B P'_c$ with $P_c = I - X(X'X)^{-1} X'$, q is the number of non-zero eigenvalues of the matrix $P_c B P_c$ and $\boldsymbol{\psi} = M \boldsymbol{\psi}^*$. The prior for the precision parameter $\tau$ can be specified as Gamma with hyperparameters chosen to be equal to 1 [11]. The prior for the vector of the fixed covariate effects is specified as

$$\boldsymbol{\beta} \sim N(\boldsymbol{\beta}^*, g^{-1} A I_2)$$

where $g$ represents the Zellner prior [39] evaluated at a fixed point estimate 10 [10]. In the prior specification for fixed effects $\boldsymbol{\beta}$ of [11], $A$ is replaced by $\tau^{-1}$.

For estimation of the fixed effects and the random effects, BEL was used in spite of having a parametric distribution for $Y_i$. The estimating equations for $(\boldsymbol{\beta}, \boldsymbol{\psi})$ are:

$$\sum_{i=1}^{n} w_i(y_i - \mu_i) = 0 \tag{5}$$

$$\sum_{i=1}^{n} \{w_i(y_i - \mu_i)^2 / \sigma_i^2\} - 1 = 0. \tag{6}$$

Details of the MCMC sampling algorithm using a random walk Metropolis-Hastings (MH) approach can be found in [11].

## 2.3 Recap of Bayesian parametric spatial models

For parametric modelling of disease incidence or mortality, the response variable is usually assumed to have a Poisson distribution with an expected value that can be explained by a function of covariates and spatial random effects. Gaussian distributions are also commonly used for modelling continuous response variables such as standardised incidence ratios (SIRs) on a logarithmic scale [40] and binomial distributions are used for proportions [41]. There is a wide range of spatial prior formulations in the literature, including basis functions, deformation methods, Gaussian Markov Random Field (GMRF) methods etc. [42]. A very popular class of priors to represent these random effects is the conditional autoregressive (CAR) models, which are a special case of the GMRF methods. Different model formulations using different CAR models are available in the literature [15, 43]. A short recap of BYM, Cressie, Leroux and generalised Moran basis priors are provided in this section.

**2.3.1 The BYM model.** A very well-known Bayesian hierarchical model for disease mapping was proposed by Besag et al. [18], known as the BYM model. The spatial random effects $\psi_i$ (from Eq 4) comprise a spatial random effect term $u_i$ with a CAR prior structure and an unstructured random component $v_i$. The conditional distribution of each $u_i$ can be expressed as:

$$u_i | u_{j, i \neq j} \sim N \left( \frac{\sum_j w_{ij} u_j}{\sum_j w_{ij}}, \frac{\sigma_u^2}{\sum_j w_{ij}} \right)$$

where $w_{ij}$ are the weights defining the relationship between area $i$ and its neighbours and the prior mean is a weighted average of the other $u_j$ [18]. A popular specification of the weights is the intrinsic CAR prior in which $w_{ij} = 1$, if area $i$ and $j$ are adjacent and $w_{ij} = 0$ otherwise. The prior for unstructured random component $v_i$ is typically considered to have an independent normal distribution,

$$v_i \sim N(0, \sigma_v^2).$$

The above model specification with both structured and unstructured random components ($u_i$ and $v_i$) with or without covariates is also known as a convolution model.

If only a structured component $u_i$ is used to express the spatial dependence (e.g. in Eq 4 $\psi_i = u_i$), then the model is termed an intrinsic model, which is the simplest possible CAR prior that does not estimate the strength of the spatial correlation between the random effects [15]. Intrinsic CAR distributions are commonly used to model the spatial dependency structure in Bayesian hierarchical models, such as the intrinsic Gaussian Markov Random Field (IGMRF) [44].

**2.3.2 Cressie model.** The Cressie Model, also referred to as a proper CAR model was proposed by Ver Hoef and Cressie [45] and Stern and Cressie [19]. The model considers a single set of random effects with an additional spatial correlation parameter $\rho$. The plausible range for $\rho$, $0 \leq \rho < 1$ makes the Cressie model a proper CAR model and becomes the ICAR model for $\rho = 1$. According to this model, the random effect $\psi$ has a multivariate normal distribution [16]:

$$\boldsymbol{\psi} \sim MVN(\mu, \sigma^2 \boldsymbol{Q}^{-1}) \tag{7}$$

where the $(i, j)$th element of $\boldsymbol{Q}$ is defined as:

$$Q_{ij} = \begin{cases} n_i, & i = j, \\ -\rho, & i \sim j, \\ 0, & \text{otherwise.} \end{cases}$$

The univariate full conditional distribution for $\psi_i|\psi_{-i}$ can be written as:

$$\psi_i|\boldsymbol{\psi}_{-i} \sim N\left(\rho\frac{1}{n_i}\sum_{j\sim i}\psi_j, \frac{\sigma^2}{n_i}\right) \tag{8}$$

where $\boldsymbol{\psi}_{-i}$ denotes the random effect vector with the $i^{th}$ component deleted. The conditional variance is the same as for the intrinsic CAR model and the conditional expectation is expressed as a weighted average of local random effects with weight $\rho$ and zero overall average weighted by $1 - \rho$. A drawback of this model pointed out by Rampaso et al. [16] is that the conditional variance depends on the number of neighbours even in the absence of spatial dependence.

**2.3.3 Leroux model.**   The Leroux spatial model, proposed by Leroux et al. [20] proposed the following distribution for the spatial random effects $\psi_i$ (from Eq 4)

$$\boldsymbol{\psi} \sim MVN(0, \boldsymbol{D}) \tag{9}$$

with a singular covariance matrix $D$. Leroux et al. [20] proposed the generalised inverse of $D$ as:

$$\boldsymbol{D}^- = \{(1 - \rho)\boldsymbol{I} + \rho\boldsymbol{R}\}/\sigma^2 \tag{10}$$

where $R$ is the intrinsic autoregression matrix which represents the neighbourhood structure of the regions with typical element,

$$R_{ij} = \begin{cases} n_i, & i = j \\ -\boldsymbol{I}(i \sim j), & i \neq j \end{cases}$$

where $n_i$ is the numbers of neighbours of region $i$ and $\boldsymbol{I}(i \sim j)$ is an indicator function taking value 1 when $i$ and $j$ are adjacent.

The term $\rho$ is introduced as a spatial dependence parameter, $\rho \in [0, 1]$, with the two extreme cases giving rise to the independence model (i.e., $\psi_i = v_i$ and $D = \sigma^2 I$, where $I$ is an identity matrix) and intrinsic autoregression (i.e., $\psi_i = u_i$ and $D = \sigma^2 R^-$). The conditional moments can be expressed as weighted averages of local moments [46]:

$$E(\psi_i|\psi_{-i}) = \frac{1 - \rho}{1 - \rho + \rho n_i} \times 0 + \frac{\rho n_i}{1 - \rho + \rho n_i}\frac{1}{n_i}\sum_{j\sim i}\psi_j \tag{11}$$

$$var(\psi_i|\psi_{-i}) = \frac{1 - \rho}{1 - \rho + \rho n_i} \times \sigma^2 + \frac{\rho n_i}{1 - \rho + \rho n_i}\frac{\sigma^2}{n_i} \tag{12}$$

where $\boldsymbol{\psi}_{-i}$ denotes the random effect vector with the $i$th element deleted. For $\rho$ close to 1, the conditional variance becomes close to $\sigma^2/n_i$ and for $\rho$ close to 0, the variance becomes close to $\sigma^2$, that is independent of the number of neighbours $n_i$ [16].

The Leroux Model [46] is more flexible than the earlier models, namely, the BYM model [18] and the Cressie model [45]. The Leroux model overcomes the shortcoming of the Cressie model which depended on the number of neighbours even when there was no spatial dependence by inclusion of the spatial dependence parameter $\rho$ which may take the value 0, thus making the variance term independent of the neighbourhood structure ($\sigma^2$). The Leroux model also captures the intrinsic autoregression specified by ICAR BYM Model when $\rho = 1$. The flexibility of the Leroux model has been one of the reasons of popularity of this model in spatial data analysis. This has also motivated the formulation of a BEL model using the Leroux

prior structure in the present study, which can take the form of the BYM model with intrinsic autoregression ($\rho = 1$) and spatially independent model ($\rho = 0$).

The univariate full conditional distribution for $\boldsymbol{\psi_i}|\boldsymbol{\psi_{-i}}$ can be written as:

$$\boldsymbol{\psi_i}|\boldsymbol{\psi_{-i}} \sim N\left(\frac{\rho}{n_i\rho + 1 - \rho}\sum_{j\sim i}\psi_j, \frac{\sigma^2}{n_i\rho + 1 - \rho}\right). \tag{13}$$

**2.3.4 Generalised Moran basis priors.** The generalised Moran basis priors are shown in section 2.2 in reference to the BEL spatial models. Hughes and Haran [14] introduced the generalised Moran basis priors within a parametric framework. The use of generalised Moran basis priors for spatial random effects is intended for dimension reduction in large spatial dataset while conducting spatial smoothing. The spatial generalised linear mixed effect models (SGLMM) [18] was extended by applying generalised Moran basis priors and was named Sparse SGLMM by Hughes and Haran [14]. The authors provided different applications using binary, count and continuous data following normal distributions to perform reasonably well in order to estimate the fixed effects more precisely with reduced dimension and computational time.

# 3 Formulation of BEL spatial model applying CAR prior structure

The approach of the present study is to extend the BEL semi-parametric models proposed by Chaudhuri and Ghosh [10] and Porter et al. [11] by applying the popular CAR prior structures (Leroux and BYM) for spatial random effects. The proposed Spatial Bayesian Empirical Likelihood Model with a CAR prior, SBEL-CAR takes two forms: a model with a BYM prior (intrinsic CAR prior) for the spatial random effects (SBEL-BYM) and a model with a Leroux CAR prior for these effects (SBEL-Leroux).

The SBEL-Leroux model is formulated below noting the special cases ($\rho = 1, 0$) give rise respectively to SBEL-BYM model with intrinsic autoregression and independent BEL model with an independent Gaussian prior for spatial random effects (SBEL-IG) proposed by Chaudhuri and Ghosh [10].

The SBEL-Leroux model utilises the parametric prior of the Leroux model for the spatial random effects, a Gaussian prior for the covariate fixed effects and EL to estimate the parameters of the SAE model (Eqs 3 and 4) without specifying the data distribution of $Y_i$. Thus, this model formulation is different from those of Porter et al. [11] and Chaudhuri and Ghosh [10] in terms of prior specification for the spatial random effects. This also required a new MCMC algorithm to obtain the posterior distributions of interest.

The empirical likelihood (EL) estimating equations utilised for estimating ($\boldsymbol{\beta}$, $\boldsymbol{\psi}$) are in Eqs (5) and (6) with $\mu_i = \boldsymbol{x_i'}\boldsymbol{\beta} + \psi_i$. The constrained optimisation of the EL estimating equations is well established [26]. Some R packages e.g., `gmm` [47] and `emplik` [48] perform the computation as well. A random walk MH sampling algorithm is proposed to fit the proposed model following the suggestions of Porter et al. [11]. These details are described below.

## 3.1 Prior distributions of SBEL-CAR models

The prior distribution of the random effect $\boldsymbol{\psi}$ is taken to be a Leroux prior given by Eq (14). Defining a precision parameter associated with the variance of the random effect as $\tau = \frac{1}{\sigma^2}$, the distribution of random effects $\boldsymbol{\psi}$ can be written as,

$$\boldsymbol{\psi} \sim MVN(\vec{0}, \tau^{-1}\boldsymbol{D}) \tag{14}$$

where $D$ is a singular covariance matrix with generalised inverse given by Eq (10). For a specific value of $\rho$ and an intrinsic autoregression matrix R, the prior density of $\psi$ can be specified as:

$$\pi(\boldsymbol{\psi}|\tau) \propto exp\left(-\frac{1}{2}\boldsymbol{\psi}'\boldsymbol{D}^-\boldsymbol{\psi}\tau\right) \tag{15}$$

The prior distribution of the fixed effects $\boldsymbol{\beta}$ can be specified following the suggestion of Chaudhuri and Ghosh [10] and Porter et al. [11] as,

$$\boldsymbol{\beta} \sim MVN(\tilde{\boldsymbol{\beta}}, g^{-1}\tau^{-1}\boldsymbol{I_p}) \tag{16}$$

where $g$ represents the Zellner prior [39], $\tau$ is the precision parameter, $p$ is the number of covariates in the study and $\boldsymbol{I_p}$ is an identity matrix of dimension $p \times p$. We specify $\tilde{\boldsymbol{\beta}} = \boldsymbol{\beta}_{WLS}$, the weighted least squares estimate of $\boldsymbol{\beta}$ following the suggestion of Porter et al. [11].

There have been many recommendations about choice of $g$ including choosing fixed scalar or introducing prior for $g$ as well [49–53]. Geinitz [49] outlined three different choices of $g$ and the intuitive interpretations of the weighting in resulting posterior from a computational perspective. The author mentioned if $g$ is chosen to be 1, it implies 50% prior weight and choice of $g$ as 10 implies 10% prior weights, whereas, increase of $g$ towards infinity leads to a diffuse prior. So, $g$ is fixed at 10 in the present study following the suggestion of Chaudhuri and Ghosh [10].

Then the prior density of $\beta$ can be written as:

$$\pi(\boldsymbol{\beta}|\tau) \propto exp\left(-\frac{1}{2}g\tau(\boldsymbol{\beta} - \boldsymbol{\beta}_{WLS})'(\boldsymbol{\beta} - \boldsymbol{\beta}_{WLS})\right). \tag{17}$$

The prior distribution of precision parameter $\tau$ can be taken as a Gamma distribution [11], i.e.,

$$\tau \sim Gamma(\alpha_1, \alpha_2). \tag{18}$$

The prior density of $\tau$ can be written as,

$$\pi(\tau) \propto \tau^{1+\alpha_1}exp\left(-\frac{\alpha_2}{\tau}\right). \tag{19}$$

### 3.2 MCMC sampling algorithm

The MCMC sampling algorithm was designed and implemented in R and is motivated by the one provided by Porter et al. [11]. Hence there are similarities in the algorithm steps with differences in some models and equations induced by the new prior specification for the spatial random effects $\psi_i$.

1. **Obtaining starting values**

   Using the gmm package [47] in R and the EL estimating Eqs (5) and (6), the maximum empirical likelihood estimates (MELE) for $\boldsymbol{\beta}$ are obtained. The initial values for $\boldsymbol{\beta}$ are chosen randomly from the prior distribution, weights $w_i$ are set to $1/n$ and $\sigma_i^2$ is replaced using the calculated residual variance for estimation purposes. Using the MELE of $\boldsymbol{\beta}$ and setting $\boldsymbol{\psi} = \vec{\mathbf{0}}$ gives the starting values of $\mu_i = \boldsymbol{x}_i'\boldsymbol{\beta} + \boldsymbol{\psi}$. The EL weights $w_i$ are calculated to satisfy

the constraint:

$$W_\mu = \left\{ \sum_{i=1}^{n} w_i = 1; w_i > 0 \ \forall \ i; \sum_{i=1}^{n} w_i m_j(y_i, \boldsymbol{\mu}) = 0 \ \forall \ j) \right\} \tag{20}$$

where $m_j(y_i, \boldsymbol{\mu})$ are the estimating equations ($j$ = 1, 2) presented in (5) and (6). This calculation can be made by constrained optimisation using `el.test` from the `emplik` [48] package in R.

2. **Sampling spatial random effects, $\boldsymbol{\psi}$**

   To sample $\boldsymbol{\psi}$, a multivariate normal proposal $\boldsymbol{\psi}^* \sim MVN(0, \Sigma)$ is used where the proposal covariance $\Sigma$ is tuned by pilot chains [54]. The proposed values are then utilised in the estimating equations below to generate weights $w_i^*$:

$$\sum_{i=1}^{n} w_i^* \{Y_i - \boldsymbol{x}_i' \boldsymbol{\beta} - \boldsymbol{\psi}_i^*\} = 0 \tag{21}$$

$$\sum_{i=1}^{n} \{w_i^* (Y_i - \boldsymbol{x}_i' \boldsymbol{\beta} - \boldsymbol{\psi}_i^*)^2 / \sigma_i^2\} - 1 = 0. \tag{22}$$

   To generate the set of weights, the MELE estimate of $\boldsymbol{\beta}$ from step 1 is used. If $w_i^*$ satisfies the constraint specified in Eq (20), a Metropolis-Hastings (MH) step is performed with the following posterior density ratio:

$$\gamma_\psi = \frac{p(Y|\psi^*, \beta) \pi_\psi(\psi^*|\tau)}{p(Y|\psi^{(t-1)}, \beta) \pi_\psi(\psi^{(t-1)}|\tau)}$$

$$\gamma_{\boldsymbol{\psi}} = \frac{\prod_{i=1}^{n} (w_i^*) exp\left(-\frac{1}{2} \boldsymbol{\psi}^{*'} \boldsymbol{D}^- \boldsymbol{\psi}^* \tau\right)}{\prod_{i=1}^{n} (w_i^{(t-1)*}) exp\left(-\frac{1}{2} \boldsymbol{\psi}^{(t-1)*} \boldsymbol{D}^- \boldsymbol{\psi}^{(t-1)*} \tau\right)} \tag{23}$$

   where $t$ = 1, 2, 3, . . . is the iteration index and $\boldsymbol{\psi}^{(t-1)^*}$ is the value of $\boldsymbol{\psi}^*$ in the previous iteration. When $t$ = 1, in the first iteration, $\boldsymbol{\psi}^{(t-1)*} = \boldsymbol{\psi}^{0*} = \vec{\mathbf{0}}$.

   To compute the ratio $\gamma_\psi$, $\boldsymbol{D}^-$ must be estimated given $\rho$. Ideally, in a parametric set up, $\rho$ is simultaneously estimated in the MCMC algorithm using an appropriate prior distribution and proposal distribution for $\rho$. A similar approach can be adapted in the semi-parametric approach as well. However, the number of parameters estimated using the SBEL approach does not leave enough information for effective sampling of $\rho$. The parameters $\psi$ and $\rho$ are strongly interdependent, which results in a very difficult posterior space for the Random Walk Metropolis-Hastings to adequately explore and obtain independent Monte Carlo draws from. In our investigations of MCMC sampling with free $\psi$ and $\rho$, we found that increases the MH kernel variance did not result in more independent samples as even a small increase resulted in a very low acceptance rate for new parameter values. It is for this reason we decided to fix one parameter and sample the other. It is to be noted that in sampling spatial random effects, use of Gibbs sampling is predominant in the literature [55, 56]. In SBEL approach, we cannot use Gibbs sampling due to lack of likelihood for the response variable. Hence, we decided to fix one parameter, $\rho$ and sample $\psi$, the spatial random effects.

   A grid search approach is recommended to find an appropriate $\rho$ using the SBEL-CAR

model structure. The posterior samples under this model can be drawn for each of the proposed values of $\rho$ and identify the most suitable $\rho$ for a given dataset can be determined using a model performance criterion, such as WAIC [57, 58]. The fixed value of $\rho$ is then used in the algorithm to sample the spatial random effects. Please see Appendix 2 of S1 File for illustration.

3. **Sampling the fixed effects, $\boldsymbol{\beta}$**

The vector $\boldsymbol{\beta}$ is sampled using a MH random walk step with a multivariate normal proposal, $\boldsymbol{\beta}^* \sim MVN(\bar{\boldsymbol{\beta}}, \Sigma_{\boldsymbol{\beta}})$ with proposal covariance tuned on the basis of pilot chains [54]. If the generated weights $w_i^*$ utilising the proposed values $\boldsymbol{\beta}^*$ verify the constraints specified in (20), a MH step is performed having posterior density ratio as:

$$\gamma_\beta = \frac{p(Y|\psi, \beta^*)\pi_\beta(\beta^*|\tau)}{p(Y|\psi^{(t-1)}, \beta^{(t-1)})\pi_\beta(\beta^{(t-1)}|\tau)}$$

$$\gamma_{\boldsymbol{\beta}} = \frac{\prod_{i=1}^n w_i^* exp\left(-\frac{1}{2}g\tau(\boldsymbol{\beta}^* - \boldsymbol{\beta}_{WLS})'(\boldsymbol{\beta}^* - \boldsymbol{\beta}_{WLS})\right)}{\prod_{i=1}^n w_i^{(t-1)*} exp\left(-\frac{1}{2}g\tau(\boldsymbol{\beta}^{*(t-1)} - \boldsymbol{\beta}_{WLS})(\boldsymbol{\beta}^{*(t-1)} - \boldsymbol{\beta}_{WLS})\right)} \tag{24}$$

where $\boldsymbol{\beta}^{*(t-1)}$ and $w_{i(t-1)}^*$ are the values of $\boldsymbol{\beta}^*$ and $w_i^*$ in the $(t-1)$th iteration respectively. For $t = 1$, $\boldsymbol{\beta}^{*(0)}$ and $w_{i(0)}^*$ are replaced by the initial estimates of $\boldsymbol{\beta}$ and weights $w_i$ are generated using the initial $\boldsymbol{\beta}$. As described above, $g$ is the Zellner prior [39]. and $\bar{\boldsymbol{\beta}}$ is the weighted least squares estimate of $\boldsymbol{\beta}$ [11].

The proposed values are accepted if $\gamma_{\boldsymbol{\beta}} > u_{\boldsymbol{\beta}}$ where $u_{\boldsymbol{\beta}} \sim Unif(0, 1)$. If Eq (20) is not satisfied, $\boldsymbol{\beta}^{(t)}$ is set equal to $\boldsymbol{\beta}^{(t-1)}$.

Notice that this step is similar to that proposed by [11]. The difference lies in using the values of $\boldsymbol{\psi}^*$ from step 2, which was estimated considering the Leroux model structure.

4. **Sampling $\tau$**

Following the suggestion of Porter et al. [11], $\tau$ is sampled with a Gaussian proposal as $\tau^{*'} \sim N(\tau, \Sigma_\tau)$, with a proposal variance tuned based on pilot chains and accepted according to a MH step with posterior density ratio as:

$$\gamma_\tau = \frac{\pi_\psi(\psi_i|\tau^*)\pi_\beta(\beta|\tau^*)\pi(\tau^*)}{\pi_\psi(\psi_i^{(t-1)}|\tau^{(t-1)})\pi_\beta(\beta^{(t-1)}|\tau^{(t-1)})\pi(\tau^{(t-1)})} \tag{25}$$

where,

$$\pi_\psi(\psi_i|\tau) = exp\left(-\frac{1}{2}\boldsymbol{\psi}_i' D^- \boldsymbol{\psi}_i \tau\right) \tag{26}$$

$$\pi_\beta(\beta|\tau) = (\boldsymbol{\beta} - \boldsymbol{\beta}_{WLS}')(\boldsymbol{\beta} - \boldsymbol{\beta}_{WLS}))g\tau \tag{27}$$

$$\pi(\tau) = \tau^{(1+\alpha_1)} exp\left(-\frac{\alpha_2}{\tau}\right) \tag{28}$$

In the numerator of the posterior density ratio in Eq (25), the current values of $\beta$, $\psi$ are utilised with the proposed values of $\tau$, $\tau^*$. In the denominator, all the values of the previous iteration are used, and $\tau^*$ is accepted if $\gamma_\tau > u_\tau$, $u_\tau \sim Unif(0, 1)$.

As per step 3, this step is also similar to that given by Porter et al. [11] with changes in the posterior density ratio due to the estimation of $\psi^*$ in step 2.

5. Steps 2–4 are repeated until convergence.
After convergence, the samples drawn for each of the parameters of interest $\psi$, $\beta$ and $\tau$ are stored as draws from the desired posterior distribution and used to obtain inferences of interest.

## 3.3 Implementation

The SBEL-CAR models (SBEL-BYM and SBEL-Leroux) were fitted in R using the MH algorithm (section 3.2). An R package called BELSpatial is made available in github (https://github.com/Farzana-Jahan/BELSpatial) that contains all the necessary code to draw posterior samples of interest from the proposed SBEL-CAR models using any areal data set. The package also contains code to draw posterior samples from the BSHEL model [11] and independent Gaussian model [10]. To implement the algorithm, the R package emplik [48] was used to calculate the EL weights corresponding to the estimating equations and the package gmm [47] was used to calculate the maximum EL estimates of the regression coefficients from the data in order to obtain the starting values of the regression coefficients in the algorithm. The initial values for the other parameters of interest were randomly generated from the respective prior distributions. Three parallel chains of 1 million iterations with a burn in period of 100,000 iterations, thinned by 10, were run to fit the models in this present study described below. The convergence of the MCMC chains was assessed by using visual diagnostics and the Gelman-Rubin diagnostic. Since no distribution is assumed for the underlying data, a larger number of iterations is needed to obtain convergence compared to an traditional Bayesian parametric spatial models. In addition to fitting the proposed SBEL-CAR models, the SBEL-IG model using independent Gaussian priors proposed by Chaudhuri and Ghosh [10] and the BSHEL model proposed by [11] were also implemented to compare the performances of the BEL spatial models following the algorithms provided by the corresponding authors. We acknowledge the existence of a Hamiltonian Monte Carlo (HMC) algorithm [38] and an R package named elhmc [59] for fitting the SBEL-IG model, but in the present study we have employed a MH algorithm instead to estimate the posteriors for all four spatial BEL models.

To compare the performance of the SBEL models with their parametric analogues, the IG model (using an independent Gaussian prior for spatial random effects), the BYM and the Leroux models were fitted using the CARBayes package [60] in R. The generalised Moran basis prior applied by the sparse SGLMM model was fitted using the R package ngspatial [61]. For comparison purposes, the number of iterations, burn in period and thinning interval were kept the same for the parametric and spatial BEL models.

## 3.4 Comparison of model performance

The posterior summaries of the parameters of interest (fixed effects and precision parameters) using the parametric and SBEL models, along with Gelman Rubin diagnostic value for convergence and WAIC [57] were used as measures of performance of the models. The WAIC, which has been used in spatial models by many authors, e.g., Aswi et al. [62] and Duncan and Mengersen [63]; a smaller value of WAIC indicates a better model fit [57]. There have been two adjustments of WAIC suggested in the literature and both are viewed as the approximations to cross validation [58]. The second adjustment to WAIC, proposed by Gelman et al. [58] gives

more stable version by computing variances for each data points and then summing. In this present study, the stable version of WAIC [58] is used.

It is to be acknowledged that there are other available model selection criteria such as K-fold cross validation, information criteria such as DIC, leave one out cross validation among others [64]. All the model selection criteria described in [64] are for non-spatial Bayesian models. These criteria are not suitable for validating a spatial model [63]. The reason of not considering any of these cross-validation techniques for model selection is that the spatial component of the model adds an additional objective that often competes with goodness-of-fit. That is, goodness-of-smoothing and goodness-of-fit criteria will often lead to different "best" models. Additionally, in a spatial setting for small area level data, leaving out one or more small areas from the model as in cross validation would completely change the neighbourhood patterns, and underlying spatial dependence. Thus the goodness of fit and model selection for spatial modelling needs different considerations [63]. So we followed the recommendations of Duncan and Mengersen [63] and chose WAIC as the model selection criteria, as it is showed valid in the context of spatial models as well. While this criteria is not perfect for spatial models, more research needs to be conducted to find out ways to apply cross validation or other selection criteria for Bayesian spatial models in small area level, which is beyond the scope of this current study.

## 4 Applications

This section presents a thorough investigation of the SBEL-CAR models. The models are applied to two well-known areal spatial datasets, namely the Scottish lip cancer data and the North Carolina Sudden Infant death syndrome (SIDS) data. The models are compared with existing parametric Bayesian spatial models and other existing SBEL models (for details, see section 3.4). The comparison with the Dirichlet process (DP) prior under BEL framework proposed by Chaudhuri and Ghosh [10] are not made in this article, considering the similar output for IG and DP models reported by these authors. A simulation study is also made to compare the performances of the parametric and semi-parametric spatial models. At last an application of proposed and existent SBEL models along with the parametric spatial models are made to very recent COVID-19 data for Europe in 2020 at the small area level.

### 4.1 The Scottish lip cancer data

The Scottish lip cancer data is a publicly available spatial dataset compiled by Kemp et al. [22] and Breslow and Clayton [65]. The Scottish lip cancer data has been analysed by many authors including Clayton and Kaldor [66] Leroux et al. [46] etc. It contains data on lip cancer incidence in males registered during the 6 years from 1975 to 1980 for 56 small areas (counties) of Scotland along with information on expected incidence and sun exposure (spatial covariate). In this project, we use this well-known data set to illustrate the proposed SBEL-CAR models.

To fit the proposed SBEL-CAR approaches with the FH model described in Eqs (3) and (4), let $Y_i$ be the log of the observed standardised incidence ratios, $logSIR$ calculated from the data by taking the ratio of observed and expected incidences in each of the 56 counties and let $\mu_i$ be the corresponding expected $logSIR$ for each county. Here, $X$ is a 56 by 2 design matrix, the first column of which corresponds to an intercept parameter and the second column of which corresponds to the measure of sun exposure in each county. The estimating Eqs (5) and (6) are used in this context to draw posterior samples from the SBEL-BYM and SBEL-Leroux models using the MH algorithm for MCMC estimation described in section 3.2. The independent Gaussian spatial model under the BEL framework (SBEL-IG) [10] and the BSHEL model [11] are also fitted in the same context in order to compare the performances. Under the parametric

setup, the following model was considered with the CAR priors (ICAR BYM and Leroux), Moran basis Priors (via Sparse SGLM) and independent Gaussian (IG) priors are employed to model the spatial random effects $\psi$:

$$Y_i | \mu_i \sim N(\mu_i, \sigma_i^2) \tag{29}$$

$$\mu_i = X' \beta + \psi_i \tag{30}$$

The parametric model is chosen to make it compatible with the estimating equations utilised to fit the spatial BEL models. Alternative choices of models are also possible for both the parametric and semi-parametric set ups, such as a Poisson model for the observed incidence utilising expected incidence as an offset variable. The estimating semi-parametric models need to be adjusted to make these comparable.

The comparative performance of the models is summarised in Table 1. It is observed that all the SBEL models provided similar estimates of posterior means for fixed effects. However, in the SBEL models, the 95% credible intervals were wider than those of the parametric models.

Hence the results obtained are encouraging as they show that the estimation for the parameters $\beta$ made from SBEL-CAR models are similar to those obtained by using parametric models. It is noted that the Scottish lip cancer example is a benchmark data set for which parametric assumptions hold, so it is not surprising to see parametric models performing

**Table 1. Posterior summaries of regression coefficients ($\beta$) and precision parameter ($\tau$) for Scottish lip cancer data using Bayesian parametric and semi-parametric models.**

|  | Models | Parameters | Mean | 95% CI | Gelman Rubin | WAIC |
|---|---|---|---|---|---|---|
| SBEL Models | SBEL-BYM | $\beta_0$ | -0.008 | (-1.65, 1.65) | 1.01 | 452.82 |
|  |  | $\beta_1$ | 0.437 | (-1.27,2.15) | 1.00 |  |
|  |  | $\tau$ | 0.035 | (0.024,0.052) | 1.00 |  |
|  | SBEL-Leroux | $\beta_0$ | -0.010 | (-1.45, 1.43) | 1.00 | 452.53 |
|  |  | $\beta_1$ | 0.437 | (-1.28,2.16) | 1.00 |  |
|  |  | $\tau$ | 0.035 | (0.024,0.052) | 1.00 |  |
|  | BSHEL | $\beta_0$ | -0.025 | (-0.89, 0.84) | 1.00 | 452.34 |
|  |  | $\beta_1$ | 0.429 | (-0.77,1.64) | 1.00 |  |
|  |  | $\tau$ | 0.392 | (0.005,3.91) | 1.00 |  |
|  | SBEL-IG | $\beta_0$ | -0.025 | (-1.31, 1.26) | 1.00 | 452.21 |
|  |  | $\beta_1$ | 0.43 | (-1.27,2.13) | 1.00 |  |
|  |  | $\tau$ | 0.035 | (0.024,0.051) | 1.00 |  |
| Parametric | BYM | $\beta_0$ | -0.014 | (-0.22,0.19) | 1.00 | 226.36 |
|  |  | $\beta_1$ | 0.276 | (-0.023,0.555) | 1.00 |  |
|  |  | $\tau$ | 0.369 | (0.018,1.58) | 1.00 |  |
|  | Leroux | $\beta_0$ | -0.013 | (-0.249, 0.221) | 1.00 | 164.96 |
|  |  | $\beta_1$ | 0.346 | (-0.030,0.663) | 1.00 |  |
|  |  | $\tau$ | 0.043 | (0.002,0.52) | 1.00 |  |
|  | Moran basis | $\beta_0$ | -0.013 | (-0.25, 0.22) | 1.00 | 163.03 |
|  |  | $\beta_1$ | 0.434 | (-0.030,0.66) | 1.00 |  |
|  |  | $\tau$ | 1.181 | (0.005,3.91) | 1.01 |  |
|  | IG | $\beta_0$ | -0.014 | (-0.27,0.23) | 1.00 | 161.37 |
|  |  | $\beta_1$ | 0.43 | (0.16,0.70) | 1.00 |  |
|  |  | $\tau$ | 0.123 | (0.002, 1.09) | 1.01 |  |

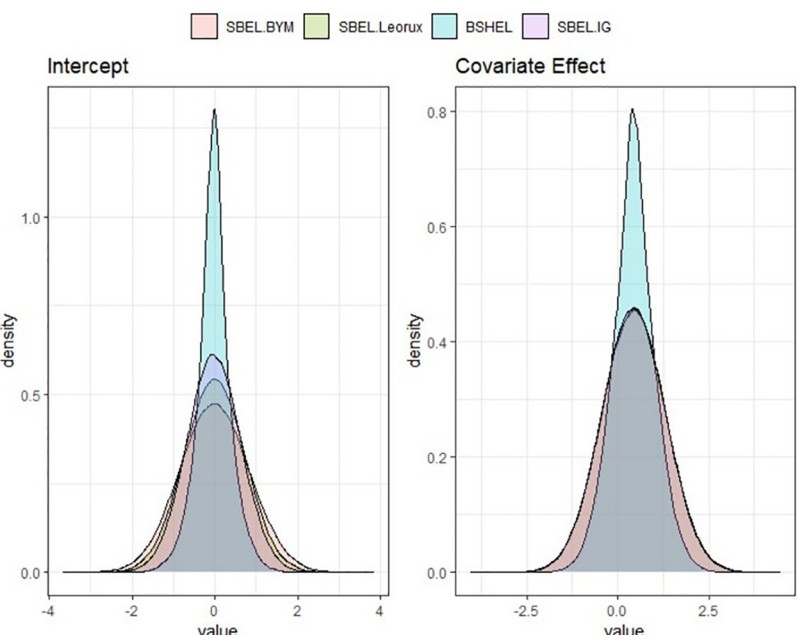

**Fig 1. Posterior densities of regression coefficients $\beta_0$ and $\beta_1$ using spatial BEL models for Scottish lip cancer data.**

better according to WAIC and posterior distributions. In cases in which it is not straightforward to assume the parametric distribution, SBEL models can be useful to draw posterior inference and make predictions.

Figs 1 and 2 show posterior densities obtained by applying the SBEL and parametric spatial models using different prior structure such as: BYM, Leroux, Moran basis prior and IG prior. It is noted that the BSHEL model outperforms the SBEL-BYM and SBEL-Leroux models based on the precision of the posterior estimates of the regression parameters, though the posterior means are all concentrated around the same value for this case study. According to the WAIC reported in Table 1, the prediction performance of the SBEL models is very similar irrespective of the choice of spatial priors. For spatial maps showing smoothed SIRs obtained by SBEL models and visualisation of posterior distribution from each of the parametric and SBEL models, see Figs A1 and A2 in S1 File.

## 4.2 The North Carolina SIDS data

The North Carolina sudden infant death syndrome (SIDS) data is also a publicly available benchmark dataset containing the annual number of SIDS and death rates per 1000 live births for each of 100 counties and each of the years between 1974–1975. The data set was first presented by Atkinson [23] and subsequent analysis and additions were made by Symons et al. [67] and Cressie [68]. The augmented version of the data is printed in Cressie, N. [69]. A more recent introduction to this data set is given by Bivand [70]. For evaluating the performance of the proposed SBEL models in this project, the aggregated counts of SIDS for 1974–78 are modelled using the corresponding spatial covariate (number of non-white births) at small area levels.

Similar to the previous case study, the proposed SBEL-CAR models are fitted considering $Y_i$ as the log of the observed standardised mortality ratios, *logSMR* calculated from the data by taking the ratio of observed and expected mortality in each of the 100 counties. Here $\mu_i$ is the expected *logSMR* for each county and $X$ is a 100 by 2 design matrix. The available covariate

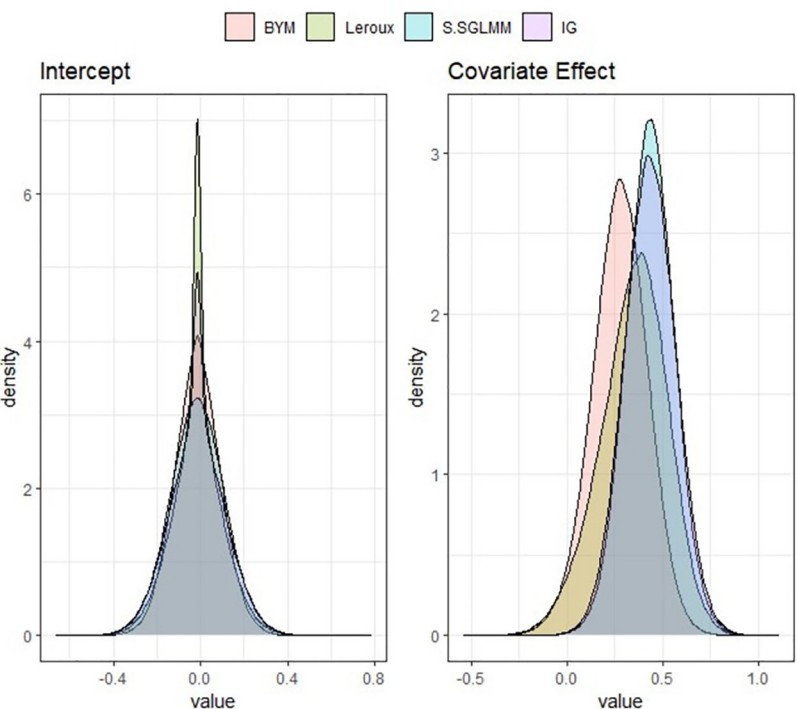

**Fig 2. Posterior densities of regression coefficients $\beta_0$ and $\beta_1$ using Bayesian parametric spatial models Scottish lip cancer data.**

information in the data set is the number of non-white births in each county, denoted by the second column in *X*. Other spatial BEL models and parametric spatial models are also fitted to compare the model performance similar to the Scottish Lip Cancer application.

The posterior estimates of regression parameters and the prediction performance of each models fitted to the data are presented in Table 2. Among the parametric models, the Leroux model shows the best performance and among the SBEL models, the BSHEL model performs the best, although the WAIC values of all the SBEL models are very close. Figs 3 and 4 show the posterior densities obtained using SBEL models and parametric models. The spatial maps showing raw and smoothed SMRs obtained by fitting BEL spatial models are shown in the Fig A4 in S1 File. All these results for this case study also suggest that the choice of spatial priors is very important when fitting Bayesian parametric spatial models using BYM, Leroux, generalised Moran basis priors or IG for spatial random effects but the performance of SBEL models is very similar irrespective of choices of spatial prior.

## 4.3 Simulated data

For a more detailed investigation of SBEL models on areal spatial data, data were generated on expected counts, based on an underlying spatial random field (USRF) following the method provided by Aswi et al. [62]. A covariate was also generated and the observed counts simulated so that they are influenced by the covariate effect as well as the USRF. The synthetic data were generated for a small (25 areas) and a large number of areas (100) for strong and weak spatial autocorrelation among the small areas. Each of the parametric Bayesian spatial models (BYM, Leroux, IG and Moran basis) and SBEL models (SBEL-BYM, SBEL- Leroux, BSHEL and SBEL-IG) were fitted to five realisations of each of the simulated data scenario taking the log of

**Table 2. Posterior summaries of regression coefficients ($\beta$) and precision parameter ($\tau$) for North Carolina SIDS data using SBEL and Bayesian parametric models.**

|  | Models | Parameters | Mean | 95% CI | Gelman Rubin | WAIC |
|---|---|---|---|---|---|---|
| SBEL Models | SBEL-BYM | $\beta_0$ | -0.326 | (-2.03, 1.37) | 1.00 | 923.01 |
|  |  | $\beta_1$ | 0.239 | (-1.48, 1.97) | 1.00 |  |
|  |  | $\tau$ | 0.018 | (0.015,0.0267) | 1.00 |  |
|  | SBEL-Leroux | $\beta_0$ | -0.310 | (-1.72,1.09) | 1.00 | 922.85 |
|  |  | $\beta_1$ | 0.239 | (-1.42,1.89) | 1.00 |  |
|  |  | $\tau$ | 0.564 | (0.015,2.52) | 1.00 |  |
|  | BSHEL | $\beta_0$ | -0.303 | (-1.07,0.48) | 1.00 | 922.44 |
|  |  | $\beta_1$ | 0.236 | (-1.039,1.53) | 1.00 |  |
|  |  | $\tau$ | 0.024 | (0.002,0.347) | 1.00 |  |
|  | SBEL-IG | $\beta_0$ | -1.01 | -0.284, 2.312 | 1.00 | 928.54 |
|  |  | $\beta_1$ | 0.121 | (-1.59,1.84) | 1.00 |  |
|  |  | $\tau$ | 0.019 | (0.015,0.026) | 1.00 |  |
| Parametric | BYM | $\beta_0$ | -0.306 | (-0.377, -0.235) | 1.00 | 485.61 |
|  |  | $\beta_1$ | 0.0325 | (-0.122,0.193) | 1.00 |  |
|  |  | $\tau$ | 2.26 | (0.446,3.239) | 1.00 |  |
|  | Leroux | $\beta_0$ | -0.306 | (-0.345,-0.267) | 1.00 | 187.85 |
|  |  | $\beta_1$ | 0.067 | (-0.094, 0.233) | 1.00 |  |
|  |  | $\tau$ | 1.63 | (0.96,2.50) | 1.00 |  |
|  | Moran basis | $\beta_0$ | -0.306 | (-0.48,-0.132) | 1.00 | 268.85 |
|  |  | $\beta_1$ | 0.238 | (0.064,0.413) | 1.00 |  |
|  |  | $\tau$ | 1.28 | (0.92,1.96) | 1.00 |  |
|  | IG | $\beta_0$ | -0.306 | (-0.477,-0.134) | 1.00 | 261.151 |
|  |  | $\beta_1$ | 0.238 | (0.0589,0.417) | 1.00 |  |
|  |  | $\tau$ | 0.010 | (0.002, 0.870) | 1.01 |  |

standardised incidence ratios, $log(SIR)$ (ratios of observed and expected disease counts) for each area as the response variable. The model performance was compared by considering the maximum values of WAIC for each simulation scenario.

An additional scenario of disease data was considered, under which the $log(SIR)$ followed a mixture distribution consisting of different Gaussian components with outliers.

The comparative performance of the models under the different simulation scenarios (Table 3) is presented in Table 4. The parametric models fitted assumes a Gaussian distribution of the response variable (Eq 29). However, the data generated for a small number of areas (25), does not necessarily reflect a Gaussian distribution (Fig 5). As a result, the SBEL models performed better than the parametric spatial models (smaller WAIC, Table 4). When the number of areas increased (to 100), the deviation from the normality assumption was reduced (Fig 5) and as a result the parametric spatial models performed better than the SBEL models. Similar results are observed for both strong and weak spatial autocorrelation. The results obtained for scenario 3, where the data contain outliers and mixture distributions (Fig 6) indicate that the SBEL models perform better than the parametric models for both small and large number of areas (smaller WAIC, Table 3).

From the simulation, it can be observed that if the underlying distribution of the response variable is irregular, contains noise or does not adhere to the parametric assumptions for the underlying distribution, applying SBEL models may be preferable in terms of model fit and precision of parameter estimates. This might arise in the case of small number of areas so that the parametric assumption is not satisfied or even in situations where the underlying areal

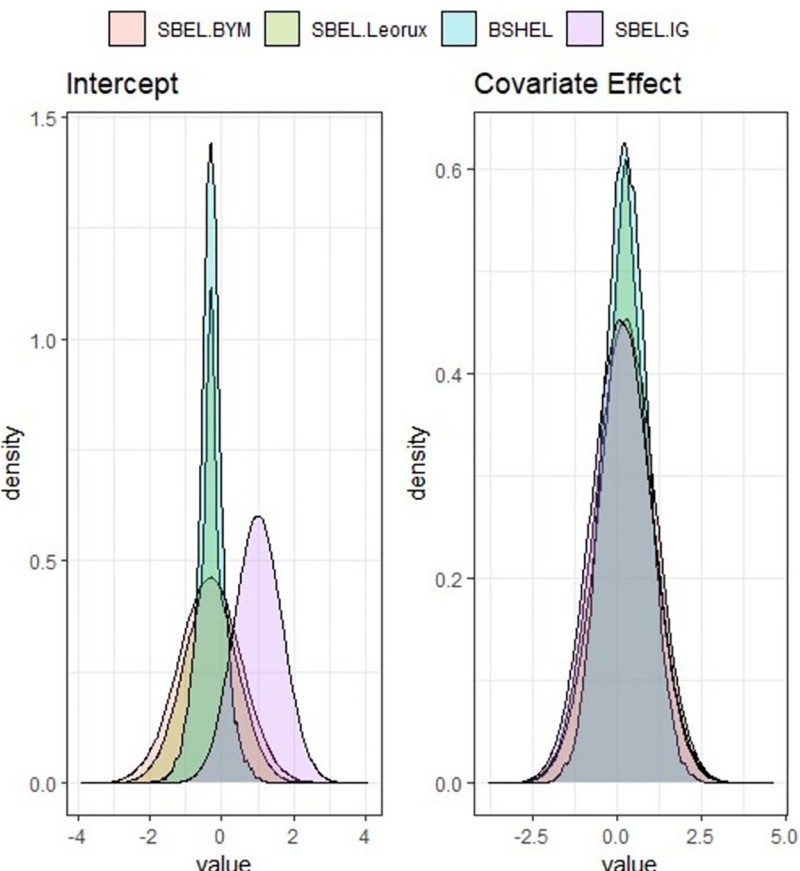

**Fig 3. Posterior densities of regression coefficients $\beta_0$ and $\beta_1$ using BEL spatial models for North Carolina SIDS data.**

spatial data comprise mixture of distributions or include extreme observations. Porter et al. [11] also shows the superior performance of BSHEL model in the presence of outliers.

## 4.4 COVID-19 data

The SBEL models and parametric spatial models were fitted to the number of deaths due to COVID-19 in the countries in Europe during three periods of 2020 (Jan-April, May-Aug and Sep-Dec). The COVID-19 data has been compiled and updated by the John Hopkins University's (JHU) Coronavirus Resource Center (https://coronavirus.jhu.edu/map.html). All the information on confirmed COVID-19 cases and deaths from the JHU data source have been combined with other information from a range of data sources such as population, number of hospital beds available and so on in "Our World in Data" website https://covid.ourworldindata.org/.

For the application of the models, the number of deaths recorded each day in each country of Europe was aggregated for each of the three periods of the year 2020 (Fig 7). The response variable is considered as the number of new deaths per million and a covariate was taken as the proportion of the population aged 65 and over (source: World Bank, compiled by "Our World in Data" website).

The variation in observed deaths per million in Europe is visible in the three time periods of 2020. To account for the temporal correlation, one option is to add a temporal component

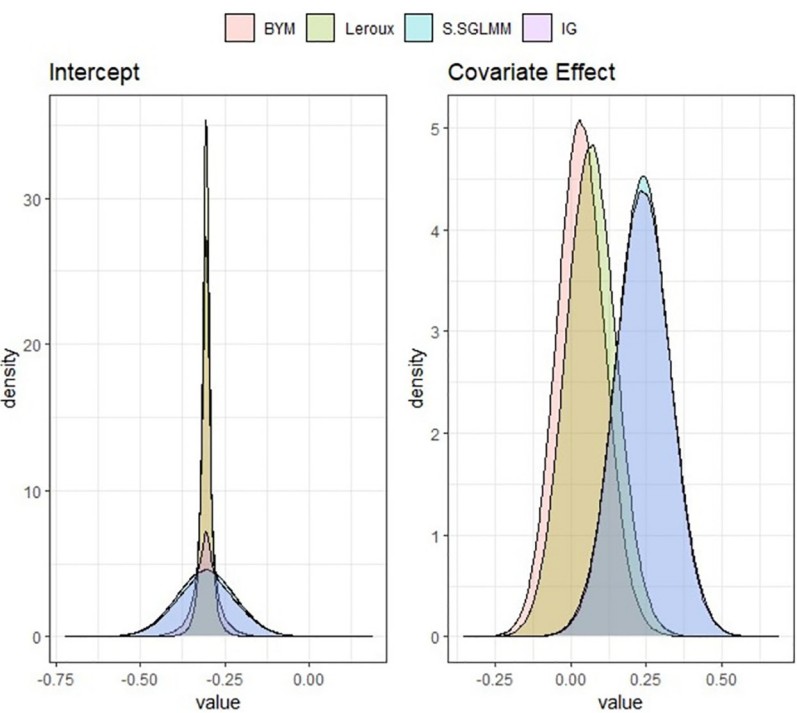

**Fig 4. Posterior densities of regression coefficients $\beta_0$ and $\beta_1$ using Bayesian parametric spatial models for North Carolina SIDS data.**

using a linear time trend, along with a spatial-temporal interaction term [71]. In the present article, we are concentrating on spatial models only and ignoring the temporal component. Adding temporal component and spatial temporal interaction might improve the model performance, which can be checked as a future extension of this study. The Fig 7 also exhibits some groups of countries having consistently lower numbers of deaths over the 3 periods, some of the countries show increase or decrease over time. More investigation is possible using the dataset in order to determine clusters and patterns of observed deaths over time, which is beyond the scope and focus of this current study.

The response variable, number of new deaths per million was log transformed in order to satisfy the required parametric assumptions (Eq 29). For the sake of comparison, $Y_i = log$(new deaths per million) is used as the response in the SBEL models as well, noting that this is not necessary and SBEL models can work without the log transformation to the response variable.

The model performance statistic, WAIC, is presented in Table 5 for each model in each time period. From this table, it is evident that SBEL models outperformed the parametric

**Table 3. Description of simulation scenarios for data generation.**

| Scenario | Description | Number of Areas |
|---|---|---|
| 1 | High autocorrelation, small N | 25 |
| | High autocorrelation, large N | 100 |
| 2 | Low autocorrelation, small N | 25 |
| | Low autocorrelation, large N | 100 |
| 1 | 20% outliers, consists mixture, small N | 25 |
| | 20% outliers, consists mixture, large N | 100 |

**Table 4. Model performance on spatial simulated data.**

| Model | WAIC | | | | | |
|---|---|---|---|---|---|---|
| | Scenario 1 | | Scenario 2 | | Scenario 3 | |
| | 25 areas | 100 areas | 25 areas | 100 areas | 25 areas | 100 areas |
| SBEL-IG | 162.29 | 922.48 | 162.3 | 924.5 | 164.8 | 922.79 |
| SBEL-BYM | 162.42 | 924.49 | 162.5 | 923.6 | 162.97 | 923.47 |
| SBEL-Leroux | 162.62 | 922.80 | 163 | 922.4 | 162.68 | 922.93 |
| BSHEL | 163.68 | 921.23 | 162.8 | 921.6 | 161.3 | 922.98 |
| IG | 260.11 | 973.66 | 205.5 | 818.6 | 518 | 924.8 |
| BYM | 407.32 | 842.36 | 214.01 | 795.6 | 263.1 | 1146.12 |
| Leroux | 210.99 | 790.22 | 207.5 | 607.3 | 263.79 | 957.3 |
| Moran basis | 189.89 | 695.91 | 246.2 | 654.5 | 274 | 964.2 |

models for all periods. It can be seen from Fig 8 that the log of new deaths per million (response variable) does not follow a normal distribution; which explains the comparatively poor performance of the parametric models which rely on this assumption. This result also supports the observations in the simulated data application about the better performance of SBEL models under situations in which the underlying parametric assumptions of the response variables are not satisfied. In terms of comparison of model performances in each period for the COVID-19 application, it is observed that the SBEL-BYM model had the smallest WAIC throughout the three periods.

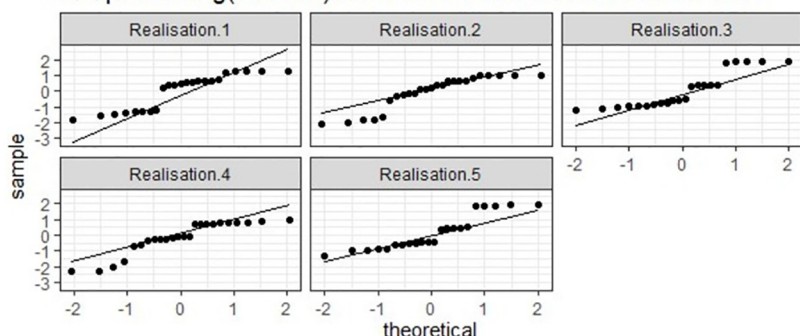

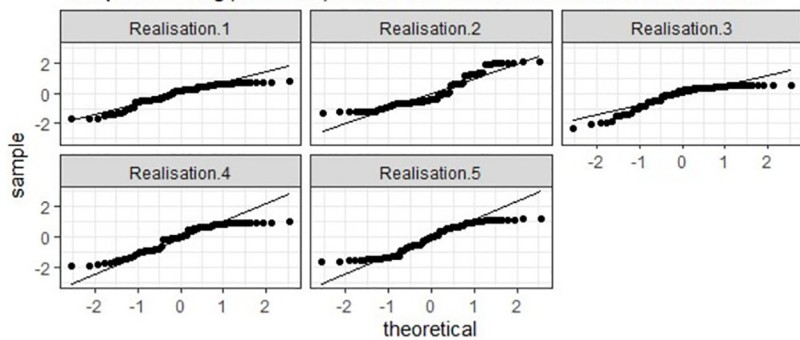

**Fig 5. The observed response variable ($y = \log(SIR)$) from simulated response under high autocorrelation (5 realisations for each of 25 areas and 100 areas).**

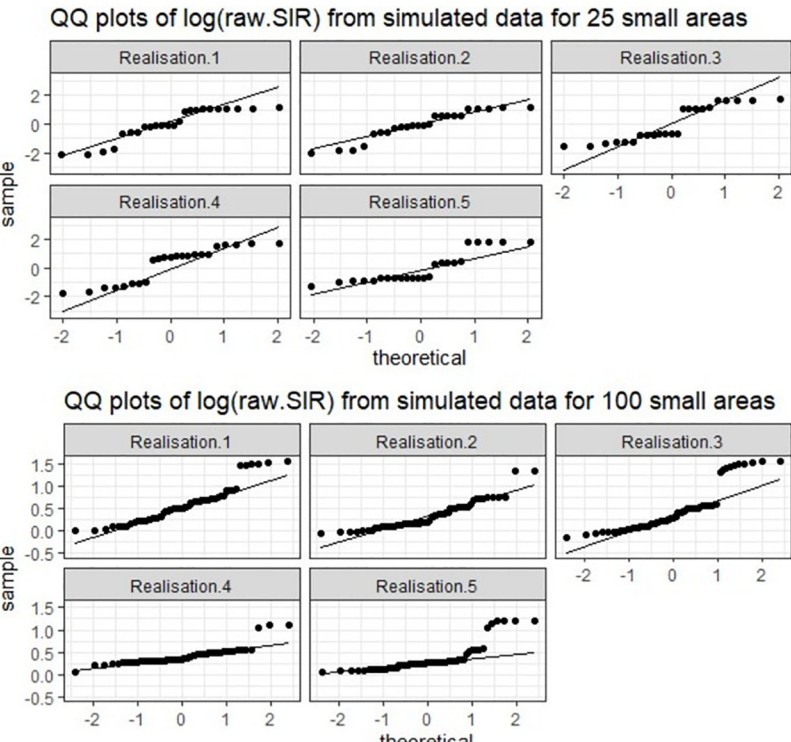

**Fig 6. The observed response variable (y = log(SIR)) from simulated response including outliers and mixture (5 realisations for 100 areas).**

The posterior distributions of the regression parameters and the precision parameters for period 1 using different models are shown using box plots in Fig 9. It can be observed that the posterior means for the regression parameters using all the parametric and SBEL models are very similar with different variability. Among all the models, BSHEL has the highest number of outliers in the posterior distribution of the regression parameters. This also resulted in the lowest mean with wider credible interval for the precision parameter ($\tau$) for the BSHEL model (Fig 9). Porter et al. [11] used three different case studies and the precision parameter behaved differently for different datasets. Hence it can be asserted that the BSHEL model is not a good choice for modelling COVID-19 death data for 2020 in Europe.

The numerical values of the posterior summaries using each model for periods 1, 2 and 3 are shown in Table A1 in Appendix 1 of S1 File. The spatially smoothed values for new deaths per million in Europe for the period 3(Sep-Dec) are shown in Fig A5 in Appendix 1 of S1 File.

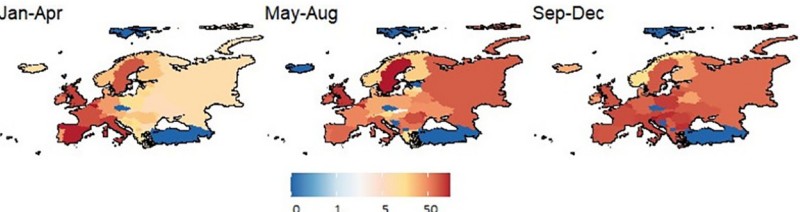

**Fig 7. Observed new deaths per million due to COVID 19 in Europe, 2020.**

**Table 5. Model performance (WAIC) for COVID-19 data models in three periods of 2020.**

| Model | WAIC | | |
|---|---|---|---|
| | Jan-Apr | May-Aug | Sep-Dec |
| SBEL BYM | 468.06 | 441.42 | 564.44 |
| BYM | 2450.72 | 4624.97 | 6169.35 |
| SBEL Leroux | 561.72 | 558.60 | 781.57 |
| Leroux | 1082.12 | 3608.08 | 6216.16 |
| BSHEL | 588.33 | 602.76 | 779.37 |
| Moran Basis | 1082.83 | 4421.82 | 2101.71 |
| SBEL IG | 534.66 | 534.68 | 758.76 |
| IG | 1081.42 | 3397.43 | 2110.83 |

## 5 Discussion

The present study investigates a Bayesian empirical likelihood (BEL) framework for modelling spatial data at small area level. The article developed spatial BEL (SBEL) models employing the popular CAR structure priors for the spatial random effects to control for the underlying spatial autocorrelation. A MCMC algorithm was proposed for SBEL-CAR models using MH techniques. A detailed comparison of the existing SBEL models (SBEL-IG and BSHEL) with the proposed SBEL-CAR models was made using two benchmark case studies employing real life spatial data at small area level, a simulation study and a study of COVID-19 deaths in Europe. A comparison with the parametric spatial models analogous to the semi-parametric models was also made in this study.

The comparison of the model performances shows that parametric spatial models are a better choice for situations in which the underlying parametric assumptions are satisfied as reflected by the model performance in the case studies using Scottish Lip cancer data and North Carolina SIDS data. However, the results of the case studies suggested that the estimated

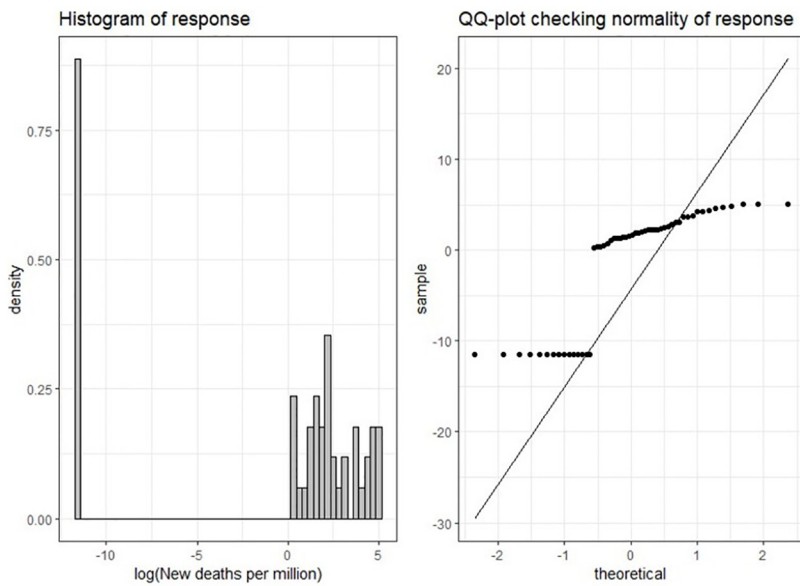

**Fig 8. Histogram and QQ plot of log(new deaths per million) in Europe during January-April, 2020.**

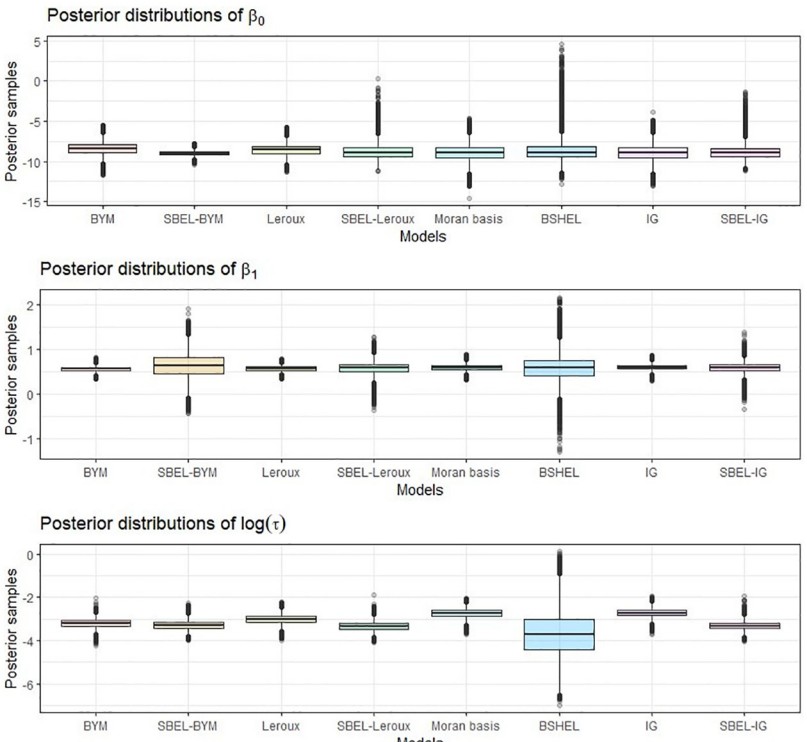

**Fig 9. Boxplots of posterior distributions of regression coefficients (parametric vs semi-parametric) for COVID-19 data for Europe, period 1 (January-April), 2020.**

regression coefficients of spatial BEL models were similar to those estimated by parametric spatial models with wider credible intervals, which is expected as the BEL models were using the empirical likelihood of the data to obtain the posterior without imposing any distributional assumptions on the data.

The simulation study revealed that spatial BEL models can outperform their corresponding parametric spatial models if the underlying assumptions regarding the data distributions are not fully satisfied. This may occur due to smaller number of areas or the presence of irregularities in the form of outliers or a mixture of different distributional components in the data. BEL has already gained popularity for providing robust estimates in the case of model misspecification [5]. Such situations might occur in the case of spatial data at a small area level, in which case SBEL models can be chosen over parametric spatial models. SBEL models have already been demonstrated to perform better than a parametric spatial model in the presence of outliers [11].

The application of the SBEL and parametric models to COVID-19 data revealed that the SBEL models performed better than the parametric models, since the distributional assumptions of the response variable were violated. This supports the finding obtained from the simulated data. This result shows that spatial BEL models could be used to analyse the very recent COVID-19 data at small area level to reveal significant trends and relationships with improved prediction performance. This is an important area of future contributions using the already existent and extended SBEL models.

This is to note that, the analysis performed with COVID-19 deaths data was not directly related to any biological insights, but it provided methods that can enable us to validate few

assumptions using the data and model outcomes. For example, from the very beginning of pandemic we are listening to the facts that COVID-19 is more severe in elderly people and immunocompromised people. In the analysis conducted in this article modelled COVID-19 deaths using the covariate proportion of population aged 65 and over in each of the 54 countries in Europe during 2020. The model revealed significant positive coefficient of the regression which in a way validated the assertion of more COVID 19 deaths on the average to the populations with higher proportion of elderly people. The proposed model was able to model the data irrespective of the irregularities and non-normality of the response variable and enabled to validate assertion with support of data. This analysis conducted is an indication of how spatial BEL models applied to COVID 19 deaths or similar data with irregularities can validate multiple research questions.

SBEL models using different priors were seen to perform similarly for benchmark datasets, while notable change in model performance is noted for Bayesian parametric spatial models with respect to the choice of spatial priors [63]. The SBEL models also showed substantive differences in the WAIC values for the COVID-19 dataset. More detailed inspection revealed that for different data sets, different SBEL models obtained the lowest WAIC. The SBEL-IG model [10] performed best for Scottish Lip cancer data analysis, while the BSHEL model [11] performed the best among the other SBEL models for the North Carolina SIDS data. For COVID-19 application, the SBEL-BYM, which is a model proposed in this study using an ICAR prior for spatial random effects within a Bayesian hierarchical modelling framework outperformed all the parametric models and the already existent BSHEL and SBEL-IG models. It is known that the performance of parametric spatial models depends on the application and data set [15], and similarly the choice of SBEL model for modelling a small area level spatial data may be decided after comparing the performance of each of the spatial BEL models for each application.

The present study has contributed to the Bayesian Empirical Likelihood literature by developing two new Bayesian semi-parametric spatial models using two popular CAR prior structures. The proposed BEL-CAR models can be extended to apply other choices of CAR priors, such as the Cressie prior [68] and the Lu prior [21] in a relatively straightforward manner. The study showed superior performance of SBEL models in situations in which parametric distributional assumptions do not hold for the data.

One limitation of the proposed SBEL models is the computational cost of the models. It requires a relatively larger number of iterations for the convergence of the MCMC algorithm and thinning is required to avoid autocorrelation. The simulation studies conducted in this paper could be extended by including more scenarios to investigate the choices of spatial models (parametric vs semi-parametric and choice of spatial priors under BEL framework) to provide more detailed insights into the choice of appropriate models to analyse areal spatial data.

## Supporting information

**S1 File. Appendices consist of all appendices, including links to download the datasets.** (PDF)

## Author Contributions

**Conceptualization:** Farzana Jahan, Kerrie L. Mengersen.

**Data curation:** Farzana Jahan.

**Formal analysis:** Farzana Jahan.

**Funding acquisition:** Kerrie L. Mengersen.

**Investigation:** Farzana Jahan.

**Methodology:** Farzana Jahan, Kerrie L. Mengersen.

**Software:** Farzana Jahan, Daniel W. Kennedy, Earl W. Duncan.

**Supervision:** Kerrie L. Mengersen.

**Validation:** Farzana Jahan, Kerrie L. Mengersen.

**Visualization:** Farzana Jahan, Earl W. Duncan.

**Writing – original draft:** Farzana Jahan.

**Writing – review & editing:** Farzana Jahan, Daniel W. Kennedy, Earl W. Duncan, Kerrie L. Mengersen.

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
