## [Decision Letter · Decision Letter 0]

6 Oct 2021

PONE-D-21-18812Evaluation of spatial Bayesian Empirical Likelihood models in analysis of small area data

PLOS ONE

Dear Dr. Jahan,

Thank you for submitting your manuscript to PLOS ONE. After careful consideration, we feel that it has merit but does not fully meet PLOS ONE’s publication criteria as it currently stands. Therefore, we invite you to submit a revised version of the manuscript that addresses the points raised during the review process.

Both referees and I have some concerns after carefully reviewing this submission, as noted below, and referee reports.

Is there any biological insight resulted from your COVID-19 data analysis?

We look forward to receiving your revised manuscript.

Kind regards,

Cathy W. S. Chen, Ph.D.

Academic Editor

PLOS ONE

Journal Requirements:

Additional Editor Comments (if provided):

Both referees and I have some concerns after carefully reviewing this submission, as noted below, and referee reports.

Is there any biological insight resulted from your COVID-19 data analysis?

Reviewers' comments:

Reviewer's Responses to Questions

**Comments to the Author**

1. Is the manuscript technically sound, and do the data support the conclusions?

Reviewer #1: Partly

Reviewer #2: No

2. Has the statistical analysis been performed appropriately and rigorously? 

Reviewer #1: Yes

Reviewer #2: No

3. Have the authors made all data underlying the findings in their manuscript fully available?

Reviewer #1: Yes

Reviewer #2: Yes

4. Is the manuscript presented in an intelligible fashion and written in standard English?

Reviewer #1: Yes

Reviewer #2: Yes

5. Review Comments to the Author

Reviewer #1: Evaluation of spatial Bayesian Empirical Likelihood model in analysis of small area data

The authors proposed Bayesian empirical likelihood models with considerations of different spatial priors to analyze the spatial datasets at small area level and provided the comparison with (semi-)parametric spatial models in this study.

Their simulation study or real data analysis demonstrated the BEL models can outperform the parametric spatial models once the assumption of distribution of data is validated. But when analyzing the data, you might consider a reasonable distribution for the data not just only assuming the normality. So I’m wondering if the performance of parametric models will be reversed (better than BEL) when you consider a reasonable model (distribution, that is Poisson, negative binomial, etc.). I admit that it makes computational complex but it could be worth some discussion.

Regarding to model selection, could they consider some other apaches for selection such as the K-fold cross-validation utility (Piironen and Vehtari, 2017) to evaluate the quality of the predictive distribution of the candidate model with the marginal predictive likelihood (MPL).

Piironen, J. and Vehtari, A. (2017). Comparison of bayesian predictive methods for model selection. Statistics and Computing, 27(3):711–735.

On page 3, I can’t see why the authors present the empirical likelihood ratio function

R(F,w)=∏_(i=1)^n▒〖nw_i 〗,

where they mentioned w_i=1/n for all i.

On page 8, could authors give more explanation why SBEL approach is not able to provide enough information to sample the spatial dependence parameter ρ using MCMC since the inter-dependent nature of ψ and ρ does not seem convincing enough?

In the section of “The Scottish Lip Cancer Data,” author compared the results from parametric and semi-parametric models and found the semi-parametric model outperforms in terms of WAIC. In parametric model, they assumed that the response follows a normal distribution, but they also mentioned that alternative model, Poisson, might be possible choice for the analysis. I’m wondering if WAIC would suggest the parametric model in this analysis once Poisson is applied here.

In the section of “Covid-19 data,” this is a very interesting analysis. Is it possible that authors can provide

the geographical distribution of some statistics from Covid-19 data like Figs A2 and A4;

the discussion how to deal with the temporal correlation for the data; and

the cluster (group, pattern) of observed data over time.

Typo: (I list some)

On page 3, line 71: Constrained optimization of the EL ratio function “is” done in order …

On page 8, line 242: a model to with …

On page 8, line 260: “Porter” et al.

On page 8, Equation (16) should be added “,” before “where g represents…”

On page 10, line 310, \\phi^*{(t-1)} is the value of \\phi^* in the previous …

Reviewer #2: see the attachment / see the attachment / see the attachment / see the attachment / see the attachment / see the attachment / see the attachment / see the attachment / see the attachment

6. PLOS authors have the option to publish the peer review history of their article (what does this mean?). If published, this will include your full peer review and any attached files.

Reviewer #1: No

Reviewer #2: No

---

## [Author Response · Author response to Decision Letter 0]

20 Dec 2021

Response to reviewers are attached as a PDF.

---

## [Decision Letter · Decision Letter 1]

7 Feb 2022

PONE-D-21-18812R1Evaluation of spatial Bayesian Empirical Likelihood models in analysis of small area dataPLOS ONE

Dear Dr. Jahan,

Thank you for submitting your manuscript to PLOS ONE. After careful consideration, we feel that it has merit but does not fully meet PLOS ONE’s publication criteria as it currently stands. Therefore, we invite you to submit a revised version of the manuscript that addresses the points raised during the review process.The authors have addressed the issues I raised. But the recaps are not well written so it is somewhat hard for us to capture the main idea of the empirical likelihood.Please submit your revised manuscript by Mar 24 2022 11:59PM. If you will need more time than this to complete your revisions, please reply to this message or contact the journal office at plosone@plos.org. Please include the following items when submitting your revised manuscript:A rebuttal letter that responds to each point raised by the academic editor and reviewer(s). You should upload this letter as a separate file labeled 'Response to Reviewers'.A marked-up copy of your manuscript that highlights changes made to the original version. You should upload this as a separate file labeled 'Revised Manuscript with Track Changes'.An unmarked version of your revised paper without tracked changes. You should upload this as a separate file labeled 'Manuscript'.If applicable, we recommend that you deposit your laboratory protocols in protocols.io to enhance the reproducibility of your results. Protocols.io assigns your protocol its own identifier (DOI) so that it can be cited independently in the future. For instructions see: https://journals.plos.org/plosone/s/submission-guidelines#loc-laboratory-protocols. Additionally, PLOS ONE offers an option for publishing peer-reviewed Lab Protocol articles, which describe protocols hosted on protocols.io. Read more information on sharing protocols at https://plos.org/protocols?utm_medium=editorial-email&utm_source=authorletters&utm_campaign=protocols.

We look forward to receiving your revised manuscript.

Kind regards,

Cathy W. S. Chen, Ph.D.

Academic Editor

PLOS ONE

Journal Requirements:

Additional Editor Comments:

The authors have addressed the issues I raised. But the recaps are not well written so it is somewhat hard for us to capture the main idea of the empirical likelihood.

Reviewers' comments:

Reviewer's Responses to Questions

**Comments to the Author**

1. If the authors have adequately addressed your comments raised in a previous round of review and you feel that this manuscript is now acceptable for publication, you may indicate that here to bypass the “Comments to the Author” section, enter your conflict of interest statement in the “Confidential to Editor” section, and submit your "Accept" recommendation.

Reviewer #1: All comments have been addressed

Reviewer #2: All comments have been addressed

2. Is the manuscript technically sound, and do the data support the conclusions?

Reviewer #1: Partly

Reviewer #2: No

3. Has the statistical analysis been performed appropriately and rigorously? 

Reviewer #1: Yes

Reviewer #2: Yes

4. Have the authors made all data underlying the findings in their manuscript fully available?

Reviewer #1: Yes

Reviewer #2: No

5. Is the manuscript presented in an intelligible fashion and written in standard English?

Reviewer #1: Yes

Reviewer #2: Yes

6. Review Comments to the Author

Reviewer #1: The authors have addressed the issues I raised. But the recaps are not well written so it is somewhat hard for us to capture the main idea of the empirical likelihood.

Reviewer #2: The authors have addressed my questions in the attached document. I have no more question. The manuscript is reasonable now.

7. PLOS authors have the option to publish the peer review history of their article (what does this mean?). If published, this will include your full peer review and any attached files.

Reviewer #1: No

Reviewer #2: No

---

## [Author Response · Author response to Decision Letter 1]

4 Apr 2022

Reviewer’s comment:

The authors have addressed the issues I raised. But the recaps are not well written so it is somewhat hard for us to capture the main idea of the empirical likelihood.

Response:

Thanks for reviewing the manuscript. In order to improve the recap of empirical likelihood, revision is made by creating two subsections on Empirical likelihood (2.1.1) and Bayesian Empirical Likelihood (2.1.2). Both these subsections now contain more information with additional references for better understanding of EL and BEL. Since there is sufficient existing literature already to explain the main idea, scope and benefits of EL, we did not elaborate this on the paper, rather we focused on using the BEL for modelling the small area level data.

---

## [Decision Letter · Decision Letter 2]

25 Apr 2022

Evaluation of spatial Bayesian Empirical Likelihood models in analysis of small area data

PONE-D-21-18812R2

Dear Dr. Jahan,

We’re pleased to inform you that your manuscript has been judged scientifically suitable for publication and will be formally accepted for publication once it meets all outstanding technical requirements.

Kind regards,

Cathy W. S. Chen, Ph.D.

Academic Editor

PLOS ONE

Additional Editor Comments (optional):

Reviewers' comments:

Reviewer's Responses to Questions

**Comments to the Author**

1. If the authors have adequately addressed your comments raised in a previous round of review and you feel that this manuscript is now acceptable for publication, you may indicate that here to bypass the “Comments to the Author” section, enter your conflict of interest statement in the “Confidential to Editor” section, and submit your "Accept" recommendation.

Reviewer #1: All comments have been addressed

2. Is the manuscript technically sound, and do the data support the conclusions?

Reviewer #1: Yes

3. Has the statistical analysis been performed appropriately and rigorously? 

Reviewer #1: Yes

4. Have the authors made all data underlying the findings in their manuscript fully available?

Reviewer #1: Yes

5. Is the manuscript presented in an intelligible fashion and written in standard English?

Reviewer #1: Yes

6. Review Comments to the Author

Reviewer #1: (No Response)

7. PLOS authors have the option to publish the peer review history of their article (what does this mean?). If published, this will include your full peer review and any attached files.

Reviewer #1: No

---

## [Editor Report · Acceptance letter]

17 May 2022

PONE-D-21-18812R2 

Evaluation of spatial Bayesian Empirical Likelihood models in analysis of small area data 

Dear Dr. Jahan:

I'm pleased to inform you that your manuscript has been deemed suitable for publication in PLOS ONE. Congratulations! Your manuscript is now with our production department. 

Kind regards, 

on behalf of

Prof. Cathy W. S. Chen 

Academic Editor

PLOS ONE